# LMAD-YOLO: A vehicle image detection algorithm for drone aerial photography based on multi-scale feature fusion

Xue Xing[1]*, Fahui Luo[1], Le Wan[1], Kang Lu[1], Yuqi Peng[1], Xiujuan Tian[2]

1 School of Information and Control Engineering, Jilin University of Chemical Technology, Jilin City, China
2 School of Transportation Science and Engineering, Jilin Jianzhu University, Changchun, China

* xingx@jlict.edu.cn

## Abstract

In the process of UAV small target vehicle detection, it is difficult to extract the features because of the small target shape of the vehicle, the environment noise is big, the vehicles are dense and easy to miss detection. The LMAD-YOLO model is proposed, and the MultiEdgeEnhancer module is designed to enhance the edge information and enhance the feature capture through a series of operations. Large Separable Kernel Attention and SPPF are combined to form MSPF module, which can realize multi-scale perception aggregation and improve the ability of distinguishing small targets from interference. Adown module is introduced to replace the model of sampling, in order to reduce the parameters and computational complexity while enhancing the accuracy of small target detection. A Multidimensional Diffusion Fusion Pyramid Network is designed, in which Dasi and feature spread mechanism are used to fuse features to reduce the error detection and missed detection. Compared with YOLO11n model P, R, MAP50 of the improved model on DroneVehicle data set were increased by 2.4%,1.4%,2.2% respectively. The model also showed good generalization ability on the VisDrone data set.

## 1. Introduction

The application of drone technology in fields such as traffic monitoring is becoming increasingly widespread, particularly demonstrating significant potential in scenarios like the collection of evidence for vehicle violations and traffic accident investigation. However, detecting small target vehicles from the aerial perspective of drones still faces numerous challenges, including low target pixel resolution, limited features, and severe environmental noise—factors that easily lead to missed or false detections. Existing deep learning-based object detection algorithms can be broadly divided into two categories. The first category is region proposal-based algorithms, such as the R-CNN series [1–3]. Although these algorithms achieve high accuracy, their complex structures result in slow detection speeds, making it difficult to meet real-time

**Data availability statement:** The data that support the findings of this study are available. The DroneVehicle data: (https://github.com/VisDrone/DroneVehicle) The VisDrone2019 data: (https://github.com/VisDrone/VisDrone-Dataset).

**Funding:** This research was funded by the Jilin Province Science and Technology Development Program [YDZJ202301ZYTS291] and the Excellent Young Project of the Jilin Provincial Department of Education for the year 2024 [JJKH20240386KJ]. The funders had no role instudy design, data collection and analysis, decision to publish, or preparation of the manuscript.

**Competing interests:** The authors have declared that no competing interests exist.

requirements—especially in drone-based real-time detection tasks. The second category is regression-based algorithms, such as the SSD series [4–6], and YOLO series [7]. These algorithms have relatively simpler structures and faster detection speeds, enabling them to balance accuracy and speed, thus having significant application value in real-time object detection.

Despite the significant progress of these algorithms, they still exhibit limitations in detecting small target vehicles from drones. First, although R-CNN series algorithms achieve high accuracy, their high computational complexity and slow inference speeds make it difficult to meet the requirements of real-time applications like drone monitoring. Second, while SSD and YOLO series algorithms offer faster detection, their limited ability to capture fine-grained features and contextual information—especially in complex aerial scenes—leads to suboptimal small-target detection performance. Third, many existing methods struggle to effectively address challenges such as environmental noise, occlusion, and high vehicle density, which are common in drone-captured images.

The detection accuracy is affected by the complex environment and high noise in small target detection.Lou et al. [8] proposed a new downsampling method to improve the feature fusion network of YOLOv8, enhancing its ability to handle dense small targets. Although this method shows some improvement, its overall performance enhancement is limited, especially in scenarios with significant background interference. Huang et al. [9] proposed DC-SPP-YOLO, which improves the Spatial Pyramid Pooling (SPP) operations to optimize feature learning and detection accuracy. By enhancing the SPP module, the model is able to capture multi-scale features more effectively, improving its ability to detect small targets in complex scenes.Bin Jiang et al. [10] enhanced the network's perception of small objects by introducing a new small-object detection layer. Specifically, they added a 160×160 detection layer to the original feature output layers of YOLOv5, fusing shallow-layer information to improve the network's sensitivity to small targets. Aiming at the mismatch between small objects and traditional anchor box sizes, they employed the K-means clustering algorithm to generate 12 scene-specific anchor boxes for the four detection layers, reducing the size deviation between anchor boxes and small objects and improving detection accuracy.Liu et al. [3] proposed a super-resolution detection framework based on high-resolution imaging. By jointly optimizing the super-resolution reconstruction and detection networks, this framework significantly improved the detection accuracy of small objects. However, the model exhibits high dependency on training data, with its performance dropping significantly when the dataset size is insufficient.

Secondly, in recent years, using multi-scale feature fusion to increase the robustness of models has become a trend.Lin et al. [11] proposed a three-scale feature fusion pyramid network, which deeply explores low-level features to enhance the detection accuracy of small targets. By leveraging multi-scale feature fusion, the model effectively captures both fine-grained details and high-level semantic information, significantly improving the detection performance of small targets in complex environments. Song et al. [12] introduced MEB-YOLO, a model

that efficiently fuses features using the Bidirectional Feature Pyramid Network (BiFPN). This architecture allows for seamless integration of features across different scales, achieving a delicate balance between computational cost and detection performance. The BiFPN-based design enhances the model's ability to detect small targets by effectively aggregating multi-scale features, which is particularly beneficial in scenarios with dense and overlapping objects. Guo et al. [13] modified the Feature Pyramid Network (FPN) to create AugFPN, which strengthens feature transfer efficiency and injects new vitality into feature processing. AugFPN introduces additional connections and attention mechanisms to enhance the flow of information between different levels of the feature pyramid, improving the model's ability to detect small targets.

Finally, in order to achieve the double optimization of accuracy and real-time in the practical application of the model, researchers have also made efforts.Chen Weibiao et al. [14] adopted a depth-separable multi-head network in YOLOv5, significantly reducing the number of parameters and model size, thereby improving computational efficiency. However, this method still faces difficulties in detecting small targets in high-noise environments.Wang et al. [15] embedded the STC structure into the neck of YOLOv8, reducing feature loss through GAM and improving detection accuracy, but this comes at the cost of a substantial increase in the number of parameters, limiting its practical application. Similarly, Wang et al. [16] optimized the backbone network and enhanced the FFNB module by integrating Biformer, expanding the detection scale and reducing the missed detection rate. However, the detection accuracy for small targets remains unsatisfactory. Zhang et al. [17] innovated a high-resolution Siamese network, incorporating an attention mechanism for learning masks. This approach achieves dual excellence in both accuracy and real-time performance, making it suitable for applications requiring high-speed detection. The attention mechanism enables the model to focus on critical regions of the image, enhancing the detection of small targets by reducing the influence of background noise. The Siamese architecture further improves the model's robustness by leveraging shared weights for feature extraction, ensuring consistent performance across different scales and resolutions.

Although the existing methods have made some progress in small target detection, they still face several key challenges when applied to UAV vehicle detection: (1) due to low resolution and noise interference, it is difficult to effectively extract small target features; (2) The ability to distinguish small targets in complex background is limited; (3) The computational complexity is high, which affects the real-time performance. These limitations highlight the need for a more robust and efficient detection model to deal with the problem of small target detection in UAV aerial scenes.

To address the above issues, the research contributions of this paper are as follows:

(1) Aiming at the problem that small targets in UAV images are limited by imaging resolution, with edge contours easily blurred by noise, leading to a decline in the model's recognition accuracy for vehicle contours and an exponential increase in the miss detection rate with noise intensity, we design the MultiEdgeEnhancer module. This module enhances vehicle edge information through average pooling and differential calculations, effectively suppressing environmental noise and improving the model's recognition and localization capabilities for small-target vehicles. Experiments show that this module increases the model's precision (P) by 2.2% and mean average precision at IoU = 0.5 (MAP50) by 0.5%.

(2) To solve the contradiction in traditional feature pyramids where shallow edge information is lost due to downsampling and deep semantic features lack local correlation during the fusion of shallow details and deep semantics, we construct the MSPF (Multi-scale Perception Fusion) module, which integrates Large Separable Kernel Attention (LSKA) and Spatial Pyramid Pooling (SPPF). By decomposing 2D large-kernel convolutions into 1D sequential convolutions of $1 \times 13$ and $13 \times 1$, this module captures long-range dependencies while reducing computational load. Combined with SPPF's multi-scale pooling, it achieves cross-scale feature aggregation, enabling dynamic interaction between shallow edge details and deep semantics in the channel dimension. This significantly improves the model's adaptability to multi-scale targets.

(3) For the issue of insufficient detection robustness in dense occlusion scenarios of UAV aerial images, we design the MDFPN (Multidimensional Diffusion Fusion Pyramid Network), introducing the Dimension-Aware Selection Integration (Dasi) module and a feature diffusion mechanism. The Dasi module dynamically allocates weights for shallow and deep features through channel partitioning, addressing the defects of traditional FPN fusion. Combined with the multi-scale feature diffusion mechanism, it promotes bidirectional feature flow across different resolution layers, enhancing the boundary discrimination ability between dense targets.

(4) To resolve the "accuracy-efficiency" dilemma caused by existing lightweight methods, which often lose small-target features due to over-simplified downsampling paths when compressing model parameters, we introduce the ADown lightweight downsampling module with a multi-path feature processing strategy. This module reduces computational load by compressing spatial dimensions through average pooling while dividing channels into two branches: one branch enhances local features via max pooling and $3 \times 3$ convolutions, and the other retains detailed information through direct $1 \times 1$ convolutions. This design reduces model parameters while avoiding feature blurring caused by traditional strided convolutions, improving the retention of small-target features under $640 \times 640$ input.

## 2. Methods and materials

### 2.1. YOLO11 object detection model

YOLO11 is the latest object detection algorithm open – sourced by the Ultralytics team in October 2024, setting a new standard in detection with an exceptional balance of speed and accuracy. The model consists of four main components: the input processing module, the backbone network, the neck network, and the prediction output head. In the input processing module, YOLO11 introduces refined data augmentation strategies, including random scaling, rotation, and color jittering, enhancing its ability to detect small objects in complex backgrounds. Additionally, it optimizes anchor box calculations and image scaling, reducing black border padding and improving computational resource utilization. The backbone network incorporates enhanced variants of cross – stage partial networks, improving the depth and breadth of feature extraction through improved feature fusion. The efficient local aggregation network module optimizes the gradient flow path, ensuring the fusion of deep and shallow features while enhancing its ability to capture complex features. The mixed pyramid convolution module combines various convolution methods to achieve multi – scale feature fusion, enriching feature representation. The neck network integrates an improved version of spatial pyramid pooling with cross – stage partial connections, capturing contextual information through multi – scale pooling and enhancing the model's adaptability to images of different resolutions. The ELAN – W module adds additional feature concatenation paths, further enhancing feature diversity and generalization capability. The prediction output head integrates multi – scale feature maps, utilizing reparameterization techniques to optimize object localization and classification, improving accuracy without increasing computational burden.

### 2.2. LMAD-YOLO

The LMAD-YOLO model is proposed to address the challenge of detecting small target vehicles in uav imagery. A multiscale feature fusion module, MultiEdgeEnhancer, is designed. The EdgeBooster sub-module first smooths the feature map via average pooling to filter noise, then highlights vehicle edges through differential calculation and convolutional activation, and finally fuses the enhanced features. The MultiEdgeEnhancer employs adaptive pooling for multi-scale feature extraction and uses EdgeBooster to enhance edges, followed by spliced fusion to improve detection accuracy. The MSPF module integrates the LSKA mechanism and SPPF.LSKA decomposes large-kernel convolutions to capture long-range dependencies, reducing resource consumption and enhancing global information modeling. Their combination enables multi-scale perception and cross-channel interaction, precisely locking onto the whole structure of small targets to enhance detection reliability.The ADown module redefines the sampling process by first using average pooling

to compress spatial dimensions and reduce computational load. It then processes channels in two branches: one applies max pooling and convolution to enhance local features, while the other uses direct convolution to preserve key details. This dual-branch design reduces parameters and computational load, thereby improving the quality of small-target features.MDFPN replaces the conventional neck network, and the Dasi module is integrated through feature dimension screening to balance the strengths of deep and shallow features. Both optimize the input features of the detection layer, reducing small-target detection errors and enhancing network performance. The network structure is shown in Fig 1.

## 2.3. MultiEdgeEnhancer module with multi-scale feature fusion

In the task of detecting small target vehicles, the vehicle targets are small, their edges are blurry, and they are highly susceptible to noise and background interference—challenges that significantly increase the difficulty of model recognition. The EdgeBooster module is designed to address these issues, and its operation process is as follows:The input feature map is first passed through an average pooling layer (AvgPool2d) to generate a smoothed feature map. Average pooling helps filter out noise, which is crucial because small vehicles have weak edges and are easily degraded by noisy signals. This smoothing step removes chaotic fluctuations, allowing subsequent processing to focus on critical vehicle edge information.The difference between the original feature map and the smoothed feature map is then computed to highlight

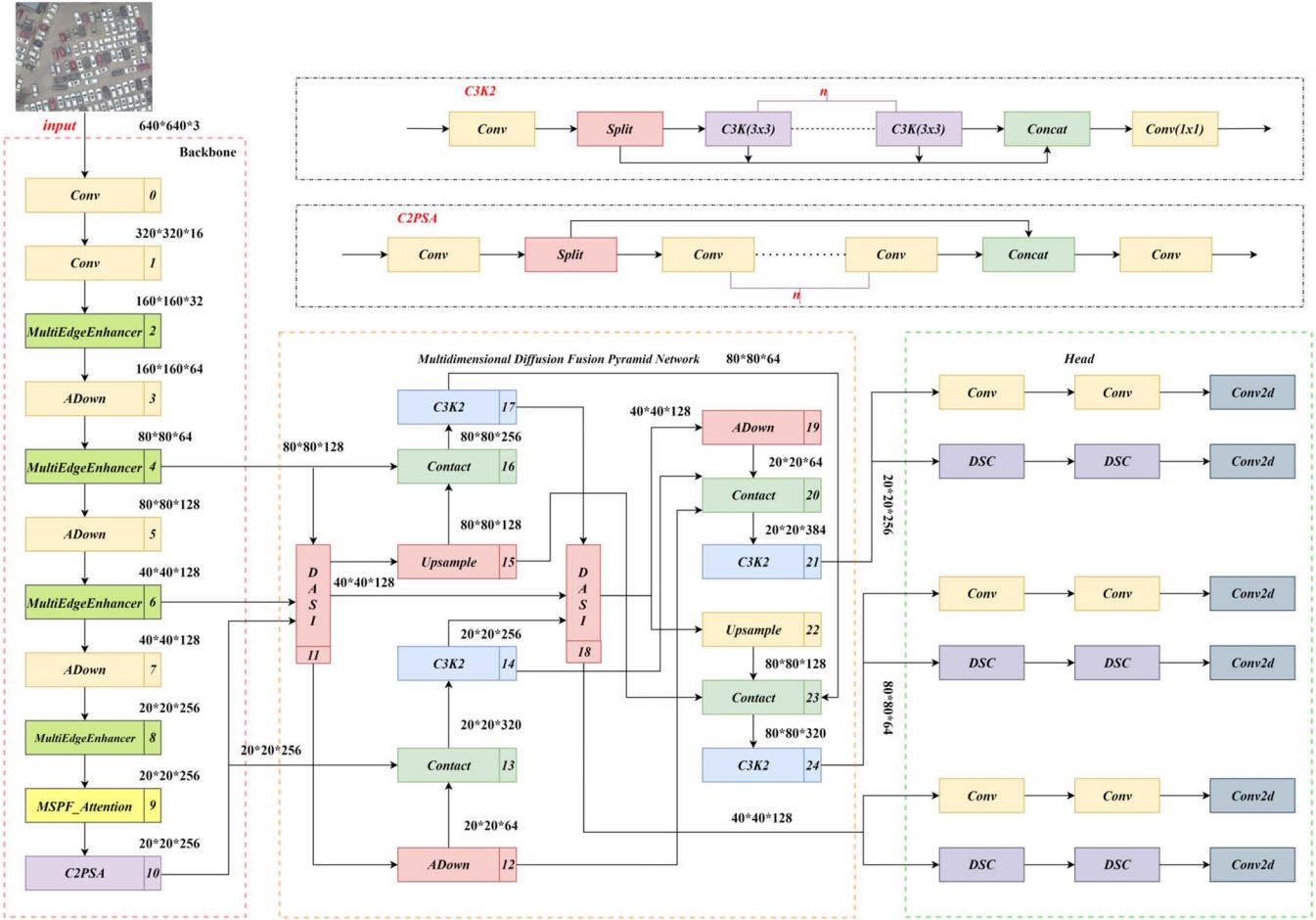

**Fig 1. Improved network structure of LMAD-YOLO.**

vehicle edge information. The edge difference map is passed through a convolutional layer for nonlinear transformation, followed by a Sigmoid activation function to capture subtle variations in vehicle edges. Finally, the enhanced edge features are added back to the original input feature map, achieving feature fusion that preserves the original context while integrating sharpened edge details.

The Adaptive Avgpool2d module achieves multi-scale feature extraction by dynamically adjusting the pooling window size. Assuming the input feature map size is $H \times W \times C$, adaptive average pooling divides the feature map into $S \times S$ sub regions based on preset scale parameters $S \in \{8, 16, 32\}$, and calculates the mean for each sub region to generate a multi-scale feature map. For example, when S = 8, the output feature map size is $\frac{H}{8} \times \frac{W}{8} \times C$. The feature maps of different scales are aligned to the original resolution through bilinear interpolation for channel concatenation, and then fused through $1 \times 1$ convolution. The formula is as follows:

$$F_{\text{fused}} = \text{Conv}\left(\text{Concat}\left(\text{AdaptiveAvgPool}_{\text{s}}\left(F_{\text{in}}\right)\right)\right), \ S \in \{8, 16, 32\}$$

(1)

The differential calculation and activation of the EdgeBooster submodule, where the input feature map $F_{in}$ first smooths the noise through average pooling to obtain the $F_{\text{smooth}} = \text{AvgPool}\left(F_{\text{in}}\right)$ residual between the original feature and the smoothed feature through differential operation:

$$F_{\text{diff}} = F_{\text{in}} - F_{\text{smooth}}$$

(2)

Subsequently, the edge response was amplified using 3x3 convolution and Sigmoid:

$$F_{\text{edge}} = \sigma\left(\text{Conv}_{3 \times 3}\left(F_{\text{diff}}\right)\right)$$

(3)

Finally, the enhanced edge features are fused with the original features in a weighted manner:

$$F_{\text{out}} = F_{\text{in}} + \gamma * F_{\text{edge}}$$

(4)

Among them are learnable weight parameters $\gamma$, with an initial value of 0.5.

We have developed a MultiEdgeEnhancer module that integrates multi-scale feature extraction, edge contour enhancement, and convolutional processing. For multi-scale feature extraction, the module employs adaptive AvgPool2d for hierarchical aggregation. Drone aerial images exhibit complex scenes with significant target size variations, where small targets are often embedded in diverse backgrounds. By defining specific scales, the module extracts local features of varying sizes and captures rich representations—ranging from fine textures to macro contours.

Edge contour enhancement is achieved via the EdgeBooster sub-module, which amplifies weak edge signals to improve the neural network's sensitivity to structural details. During feature fusion, the module first aligns multi-scale features to a unified resolution using interpolation, concatenates them into a comprehensive feature map, and then performs deep fusion via convolutional layers to generate a unified feature representation. This design integrates contextual, structural, and scale-specific information, enhancing scene perception and enabling precise capture of critical features. Ablation experiments show that compared to YOLO11n, the module boosts mAP@50 by 0.5% and precision by 2.2%. The module's architecture is illustrated in Fig 2.

## 2.4. MSPF module of LSKA attention mechanism is introduced

In UAV aerial small-target detection, the task faces severe challenges from environmental complexity and numerous interference factors: targets with low template overlap and color similarity to the background are often missed. Thus, detection models must possess both strong feature extraction capabilities and the ability to accurately distinguish targets from

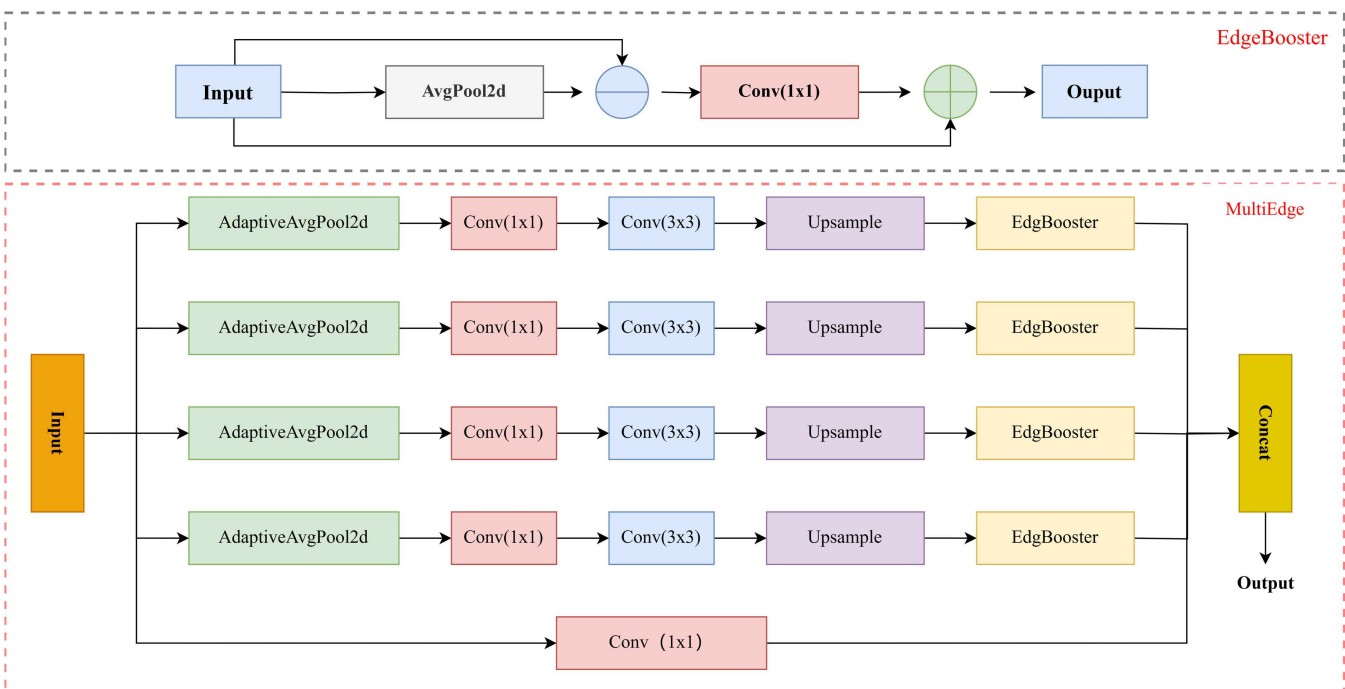

**Fig 2. MultiEdgeEnhancer module and submodule EdgeBooster.**

interfering objects in complex environments, ensuring detection accuracy and robustness.The traditional Spatial Pyramid Pooling Fusion (SPPF) module is effective in feature extraction, but its fixed-size pooling operations limit the capture of large-range features and lack global context modeling. These limitations are particularly acute in UAV small-target detection, where small targets heavily depend on global context for accurate identification.

To address this, the LMAD-YOLO model incorporates the Large Separable Kernel Attention (LSKA) mechanism [18]. LSKA captures long-range dependencies by decomposing two-dimensional large-kernel convolutions into series of one-dimensional kernels, reducing computational complexity and memory requirements while significantly enhancing global context modeling for UAV small targets. When combined with SPPF to form the MSPF module, this design maintains multi-scale feature fusion while improving feature representation. After concatenation, cross-channel interactions via LSKA and convolutions efficiently aggregate parallel branch features, enabling accurate capture of small targets' holistic structures and enriching multi-scale information fusion between shallow and deep layers. This enhances the model's ability to process image features, distinguish targets from similar interferers, and improve detection accuracy.

LSKA decomposes large kernel convolution into one-dimensional sequence operations to reduce computational complexity. Taking the 13 * 13 convolution kernel as an example, decompose it into horizontal 1 * 13 convolution and vertical 13 * 1 convolution:

$$F_{horiz} = Conv_{1\times13}(F_{in}), \quad F_{vert} = Conv_{13\times1}(F_{horiz}) \tag{5}$$

Capture long-range dependencies through concatenation operation, with a parameter count of 1/13 of traditional 2D convolution. Subsequently, a channel attention mechanism is introduced to generate a weight matrix $W_{attn} \in R^{C\times1\times1}$:

$$W_{attn} = Sigmoid(MLP(GAP(F_{vert}))) \tag{6}$$

The final output is:

$$F_{LSKA} = F_{vert} \otimes W_{attn} \tag{7}$$

In the MSPF module, LSKA and SPPF process feature maps in parallel. SPPF extracts local contextual information through multi-scale pooling, outputs features and concatenates them with LSKA results through convolution fusion.The structure of the MSPF module is shown in Fig 3.

## 2.5. Adown module with improved sampling

To address the issue of feature information loss caused by conventional downsampling methods in UAV-based small target vehicle detection, this study introduces a lightweight Adown module [19]. The proposed module employs a multi-path processing strategy to enhance feature representation: initially reducing feature map spatial dimensions through average pooling to decrease computational load, followed by channel-wise partition into dual processing streams. The local feature branch combines max pooling for enhanced feature response with stride-free convolution, while the global branch employs standard convolution operations. This dual-branch architecture effectively preserves critical feature details while maintaining low parameter count and computational complexity. The Adown module balances feature preservation and computational efficiency through a dual branch structure

  Branch 1 (Local Feature Enhancement): Perform 2 * 2 max pooling dimensionality reduction on the input feature map $F_{in}$, and then extract local features through 3 * 3 stride free convolution:

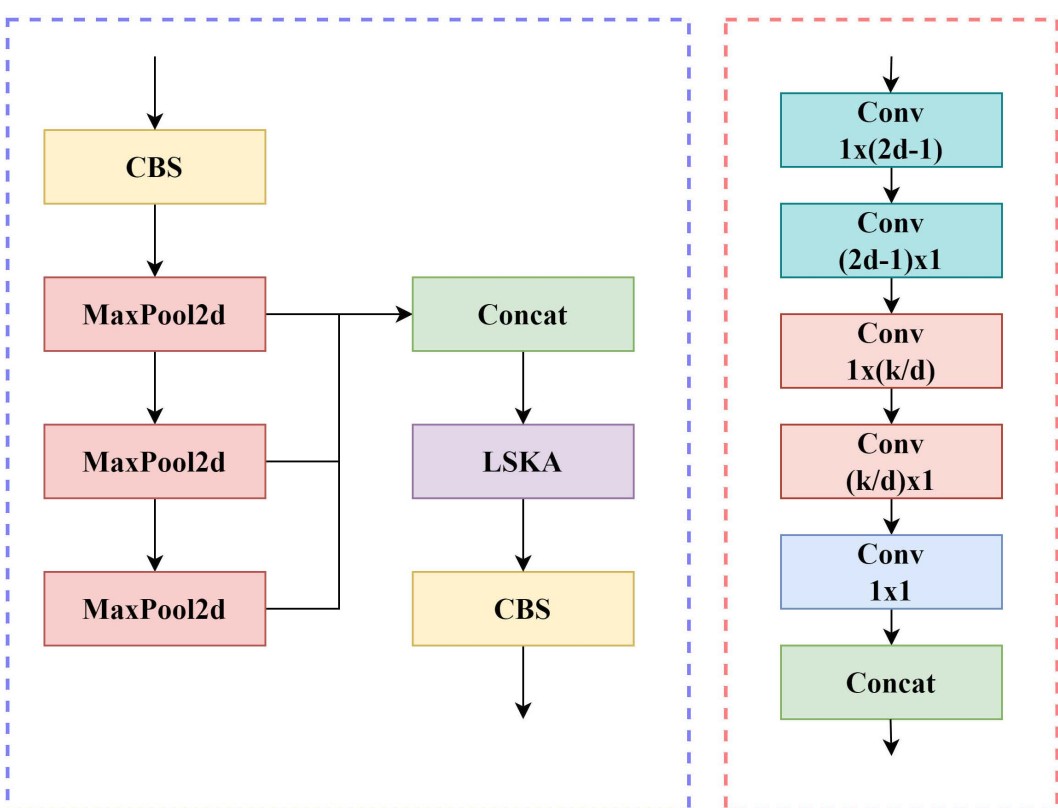

**Fig 3. MSPF and SPPF structure diagram.**

$$F_{branch1} = Conv_{3\times3}(MaxPool2_{2\times2}(F_{in}))  \tag{8}$$

Branch 2 (detail preservation): Directly perform 1 * 1 convolution dimensionality reduction on the input feature map:

$$F_{branch2} = Conv_{1\times1}(F_{in}) \tag{9}$$

The final output is a channel concatenation of two branch results:

$$F_{out} = Concat(F_{branch1}, F_{branch2}) \tag{10}$$

This design reduces the number of parameters while avoiding feature blurring caused by traditional stride convolution.

Compared to the strided convolution downsampling in YOLO11, the Adown module significantly mitigates small target feature degradation and improves feature map quality. In the LMAD-YOLO framework, traditional downsampling modules in both backbone and feature fusion networks are replaced with Adown modules, with the detailed architecture illustrated in Fig 4.

## 2.6. Multi-dimensional diffusion fusion pyramid network

In the field of target detection, the traditional Feature Pyramid Network (FPN) algorithm often falls into the predicament of high false – positive and false – negative rates when facing the small – target detection task. In view of this, a Multidimensional Diffusion Fusion Pyramid Network (MDFPN) is designed to replace the neck network in YOLO11 and improve the fusion efficiency of feature information.The network innovatively introduces the dimension - aware selection integration

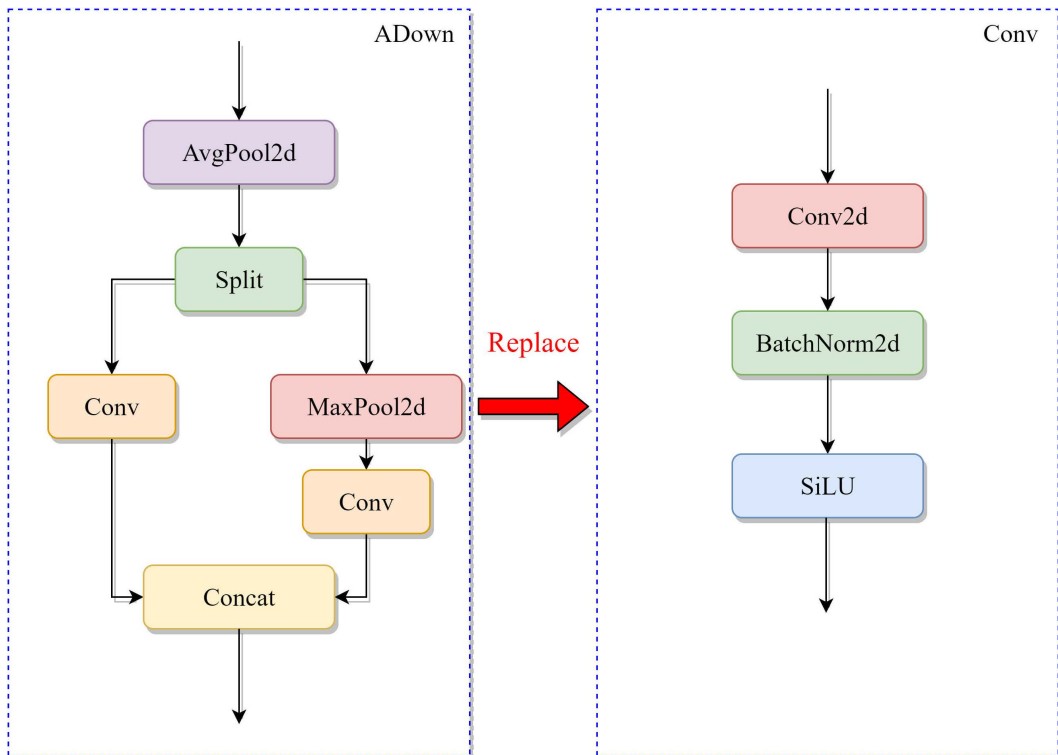

**Fig 4. ADown module replaces Conv downsampling.**

module (Dasi) and uses a multi – scale feature spreading mechanism. Dasi can intelligently select and integrate the most valuable features according to the features of different dimensions, so that the features of each scale can obtain rich and detailed context information. The spreading mechanism of multi – scale features further promotes the spread and fusion of features in the network, making the correlation between features stronger. This lays a solid foundation for the subsequent work of target detection and classification.

By effectively fusing multi – scale features, the network has greatly improved its performance in target detection or segmentation tasks. Especially in small – target detection, the network adopts the adaptive feature – selection technology, which can automatically adjust the feature – selection strategy according to the unique features of small targets. Combined with the dimension – perception mechanism, the network can accurately capture the key features of small targets, greatly enhancing the significance of small targets in the whole detection process and effectively reducing false detections and missed detections.In the process of small – target detection, deep – layer features may lose the detailed information of small targets as the down – sampling progresses, while shallow – layer features retain more details but often lack sufficient contextual information for accurate detection. To effectively balance the contradiction between deep – layer and shallow – layer features, Xu S et al. [20] proposed a channel partition selection mechanism, Dasi. See Fig 5 for its structure diagram.

The Dasi module first makes use of preprocessing operations such as convolution and interpolation to align the deep features and shallow features with the current – layer features to ensure their consistency in the spatial dimension. Subsequently, the aligned features are evenly divided into four sections in the channel dimension, and each section represents the deep, shallow, and current – layer features, respectively. This partitioning process is based on the following formula.

$$\alpha = \text{sigmoid}(u_i), \ u'_i = al_i + (1-\alpha)h_i \tag{11}$$

$$F'_u = [u'_1, u'_2, u'_3, u'_4], F_u = \delta\left(\left(\text{Conv}\left(F'_u\right)\right)\right) \tag{12}$$

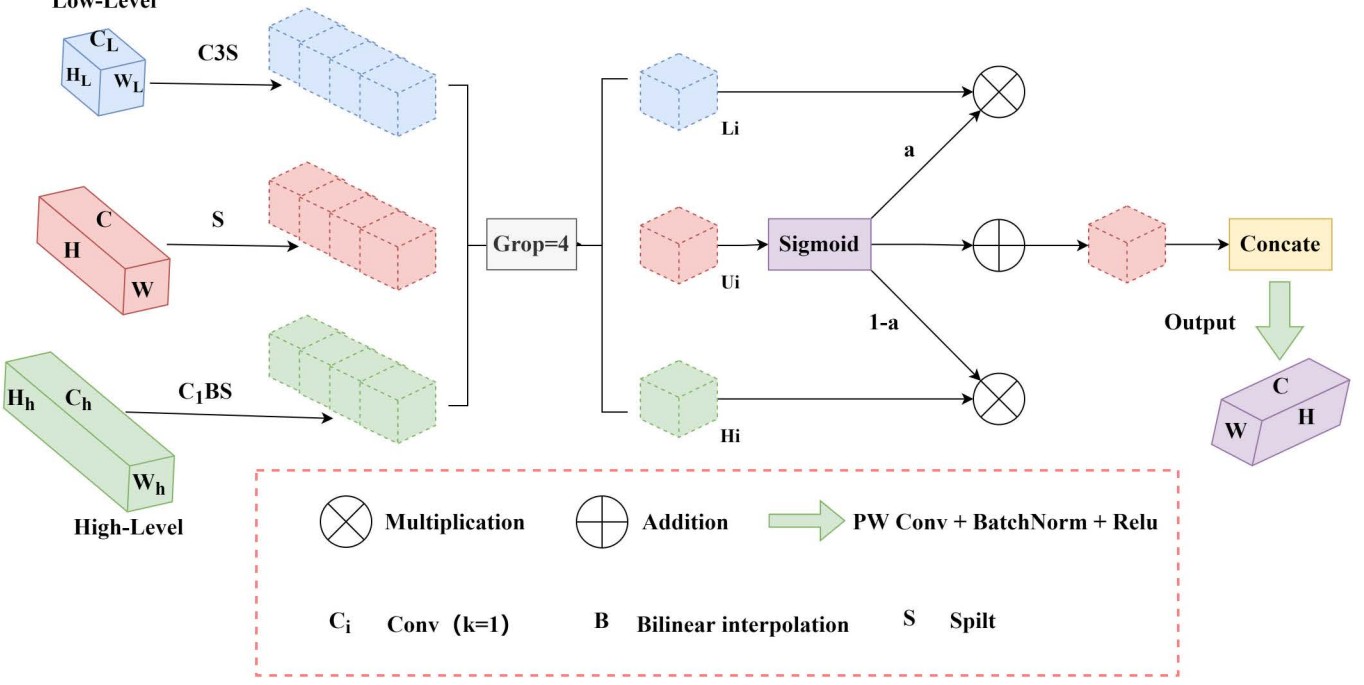

**Fig 5. DASI process diagram.**

The formula obtains a weight matrix through the calculation of the activation function, which reflects the importance of the characteristics of each partition. Then, using this weight matrix, the DASI module selectively aggregates the features of each partition to get the output of each partition. A new feature map is obtained, which combines the features of high – dimension, low – dimension, and the current layer. Then, the new feature map is further refined by convolution (Conv()), batch normalization (B()), and rectified linear unit (ReLU()), and the output feature map is finally obtained. The value of the weight matrix plays an important role in feature selection and fusion. When the value is greater than 0.5, it means that the model is more inclined to select fine – grained features that provide more detail; when the value is less than 0.5, it shows that the model pays more attention to the fusion of contextual features, i.e., those features that can provide the overall background information. The ability of adaptive feature selection based on target size and features enables the Dasi module to perform excellently in small – target detection tasks.

In the field of target detection, the efficient use of features is of great significance for improving detection performance. The multi – scale feature spreading mechanism of the multi - dimensional diffusion fusion pyramid network (MDFPN) provides an effective solution for this problem. The process is as follows: First, for feature input and initial processing, P3, P4, P5 are connected to the input feature focus module DASI. Then, in the feature – processing operation, the features output from DASI undergo C3K2 convolution and down – sampling by the Adown module. The processed features are then fed into the C3K2 module by connecting them to the P3 and P5 modules. After feature fusion through the Contact module, the processes of convolution, down – sampling, and feature connection and fusion are repeated. Through multiple cycles of up – sampling, down – sampling, and feature fusion operations in the UPSAMPLE module, the features are diffused among different scales. Finally, the Detect layer uses the features optimized by the multi – scale feature spreading mechanism to detect and identify objects. The whole network architecture and processing flow aim to provide accurate features for the Detect layer. The multi – scale feature spreading mechanism has obvious advantages. It can effectively integrate multi – scale features, allowing different – scale features to complement each other. This mechanism can obtain information from other scales to enhance the small – target feature representation, reduce false detections and missed detections, and improve the performance of the network.

## 2.7. Experimental evaluation indicators

The detection accuracy is evaluated using the mean average precision (mAP). Specifically, mAP@0.5 refers to the average precision calculated for all detection categories when the Intersection over Union (IoU) threshold is set at 0.5; whereas mAP@0.5:0.95 indicates the comprehensive calculation results of average precision for all detection categories across different IoU threshold values ranging from 0.5 to 0.95 with a step size of 0.05. The computational load of the model is measured in terms of the number of floating-point operations (GFLOPS). The GFLOPS value reflects the magnitude of floating-point operations involved during the execution of the model; a higher value indicates that the operations required during computation are more complex, leading to higher demands on computational resources. The spatial complexity of the model is assessed through the number of model parameters. The parameter count represents the total number of trainable parameters contained within the model; a larger parameter count implies greater storage space requirements for the model and also reflects the model's complexity to some extent. The formulas for calculating each metric are as follows:

$$P = \frac{TP}{TP + FP} \tag{13}$$

$$R = \frac{TP}{TP + FN} \tag{14}$$

$$\text{m}AP = \frac{1}{n}\sum_{i=1}^{n} AP_i \tag{15}$$

$$\text{m}AP@0.5 = \frac{1}{n}\sum_{i=1}^{n} AP_i(IoU = 0.5) \tag{16}$$

TP refers to samples that are positive in both actual and predicted cases; FP refers to samples that are actually negative but misclassified as positive; TN refers to samples that are negative in both actual and predicted cases; FN refers to samples that are actually positive but misclassified as negative.

### 2.8. Data set preparation

The Drone Vehicle Detection Dataset [21], collected and published by Tianjin University's research team, was used to comprehensively evaluate the performance of the improved LMAD-YOLO model. The dataset contains 28,439 RGB images and 452,570 annotations for vehicle detection, divided into five detailed categories: the common "Car" (389,779 annotations), "Truck" (22,123 annotations), "Bus" (15,333 annotations), "Van" (11,935 annotations), and the specialized "Freight Car" (13,400 annotations).

In the original dataset, a 100-pixel-wide white border was added around each image to handle object labeling at the image boundary. However, these borders provide no practical value for model training and testing and introduce unnecessary noise. Therefore, in this study, we applied border cropping to remove the white edges and correspondingly adjusted the labeling file. A subset of the dataset's images is shown in Fig 6.To validate the improved model's generalization performance, we also evaluated it on VisDrone2019 [22]. This dataset includes 10 types of aerial targets, with 6,471 training images, 548 validation images, and 1,610 test images.

### 2.9. Setting of experimental environment and training parameters

The GPU used in the experiment was Nvidia GeForce RTX 3090 with 24G of RAM, the CPU model Intel (R) Xeon (r) Gold 6134 CPU@3.20GHz, Python version 3.8.18, pytoch framework version 1.10.1+CU111, CUDA version 11.1, the training parameters are shown in Table 1.

## 3. Results

### 3.1. Ablation experiment verification

To evaluate the effectiveness of the proposed strategy in improving the detection efficiency of small targets in UAV images, an ablation experiment was conducted based on the YOLO11 algorithm, systematically analyzing the performance impacts of individual and combined modules. The experimental configurations and results are detailed in Table 2, with Group 1 serving as the baseline (YOLO11n model) and Groups 2–8 incorporating different combinations of the proposed modules.

**3.1.1. Individual module contributions.** MultiEdgeEnhancer Modul(Group2 vs. Group1):Introducingthe module improved precision (P) from 74.6% to 76.8% and mAP50 from 71.4% to 72.0%. This demonstrates its effectiveness in enhancing edge feature extraction and noise suppression, thereby improving the model's recognition and localization capabilities for small targets.

MSPF Module (Group3 vs. Group1):The MSPF module, integrating the LSKA mechanism and SPPF, maintained the baseline mAP50 (71.4%) but increased recall (R) from 66.5% to 67.0%. This indicates its role in enhancing global contextual modeling and distinguishing small targets from complex backgrounds through long-range dependency capture.

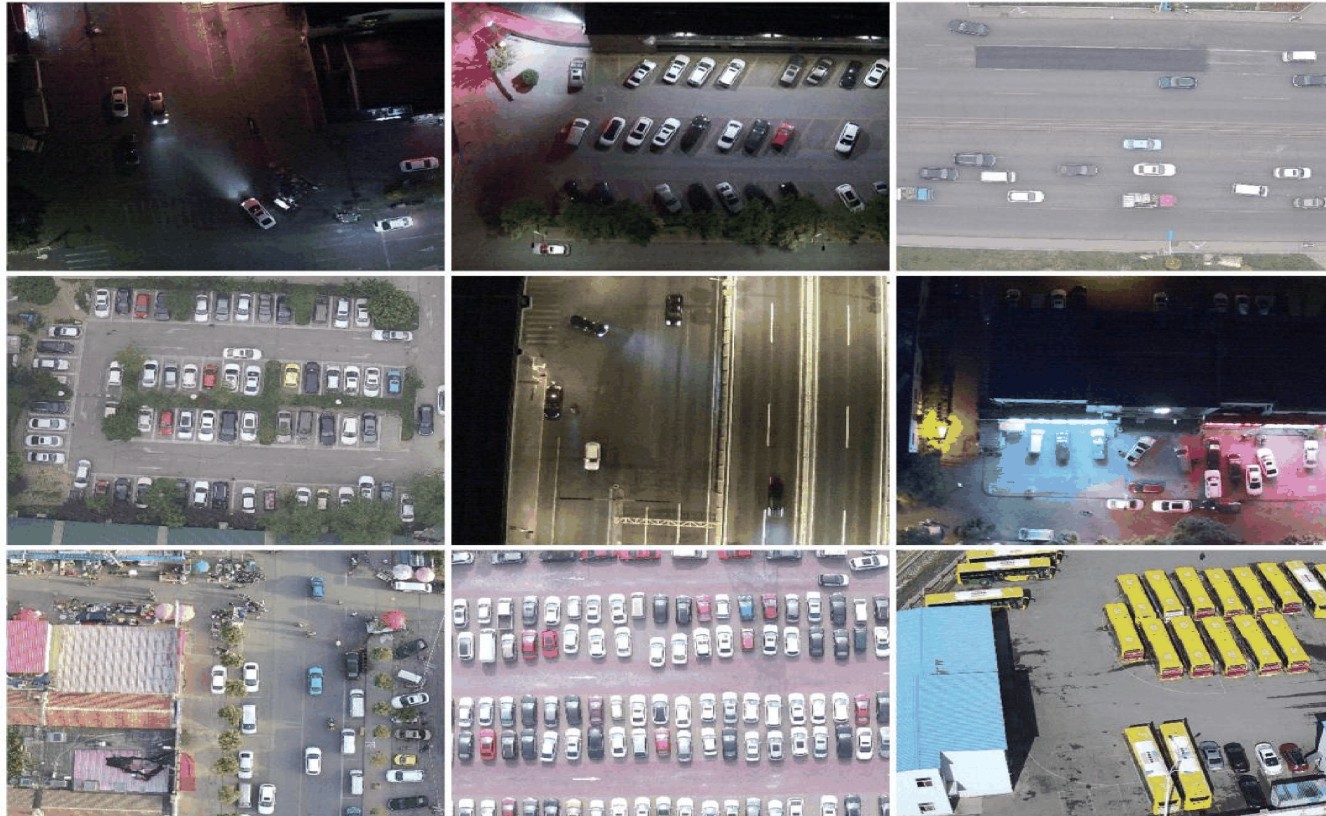

**Fig 6. Partial dataset images of DroneVehicle.**

**Table 1. Setting of training parameters.**

| Training parameters | Settings |
|---|---|
| Epochs | 200 |
| Batch | 32 |
| Imgsz | 640 |
| Workers | 4 |
| Optimizer | SGD |
| Cache | False |

**Table 2. Results of ablation experiment.**

| Group | Multie | MSPF | Adown | MDFPN | P/% | R/% | Map50/% | Map50-95/% | Params/m | GFLOPS |
|---|---|---|---|---|---|---|---|---|---|---|
| 1 | × | × | × | × | 74.6 | 66.5 | 71.4 | 45.2 | 2.583 | 6.3 |
| 2 | √ | × | × | × | 76.8 | 66.0 | 72.0 | 45.7 | 2.530 | 6.3 |
| 3 | × | √ | × | × | 73.7 | 67.0 | 71.4 | 44.9 | 2.856 | 6.5 |
| 4 | × | × | √ | × | 74.5 | **68.6** | 72.9 | **47.4** | **2.103** | **5.3** |
| 5 | × | × | × | √ | 76.1 | 68.2 | 73.4 | 47.3 | 2.680 | 7.4 |
| 6 | √ | √ | × | × | 75.2 | 66.1 | 71.6 | 45.5 | 2.804 | 6.5 |
| 7 | √ | √ | √ | × | 76.0 | 67.4 | 73.2 | 46.7 | 2.324 | 5.5 |
| 8 | √ | √ | √ | √ | **77.0** | 67.9 | **73.6** | 46.7 | 2.422 | 6.6 |

ADown Module (Group4 vs. Group1):Replacing traditional downsampling with the ADown module reduced parameters by 18.5% and GFLOPS by 15.8% while increasing R by 1.9% and mAP50 by 1.5% (to 72.9%). This validates its ability to preserve small-target features and reduce computational complexity without sacrificing accuracy.

MDFPN Network (Group5 vs. Group1):The MDFPN network, featuring the Dasi module and multi-scale feature diffusion, increased P, R, and mAP50 by 1.5%, 1.7%, and 2.0% (to 76.1%, 68.2%, and 73.4%, respectively), despite a slight rise in computational load. This highlights its effectiveness in balancing shallow detail and deep semantic features to reduce misdetections and false positives.

### 3.1.2. Module synergies and complementarity.
Edge Enhancement and Global Attention (Group6 vs. Group2):While the MultiEdgeEnhancer module alone slightly reduced recall (R from 66.5% to 66.0% in Group 2 due to potential noise amplification), integrating it with the MSPF module (Group 6) restored R to 67.4%. The LSKA mechanism in MSPF suppressed background noise through global context modeling, demonstrating a synergistic balance between edge clarity and noise resilience.

Lightweight Downsampling and Semantic Fusion (Group8 vs. Group4):The ADown module's aggressive spatial compression in Group4 caused a 1.8% decline in mAP50-95 (semantic-rich metric), reflecting potential loss of deep features. Coupling ADown with MDFPN (Group8) mitigated this issue: the Dasi module dynamically fused shallow details and deep semantics, recovering mAP50-95 to 46.7% while maintaining parameter efficiency (2.422M params, 6.6 GFLOPS). This highlights MDFPN's role in compensating for semantic degradation from lightweight downsampling.

Joint Optimization of Four Modules (Group8):The full integration of MultiEdgeEnhancer, MSPF, ADown, and MDFPN achieved the best overall performance: P＝77.0%, R＝67.9%, mAP50＝73.6%, surpassing the best single-module configuration (Group5, mAP50＝73.4%) by 0.2%. Notably, this was achieved with a minimal increase in parameters (2.422M vs. baseline 2.583M) and GFLOPS (6.6 vs. baseline 6.3), verifying positive synergies among modules.

### 3.1.3. Visualization of training dynamics.
Fig 7 visualizes the training curves of LMAD-YOLO and the original YOLO11n model. The yellow curve (LMAD-YOLO) consistently outperforms the blue curve (YOLO11n) across all epochs in precision, recall, mAP50, and mAP50-95, visually confirming the improved model's superior convergence and robustness throughout training.

The ablation experiments validate the individual effectiveness of each proposed module and their synergistic improvements, demonstrating that the LMAD-YOLO model achieves a balanced optimization of accuracy, efficiency, and feature representation for UAV small-target vehicle detection.

## 3.2. Feature map visualization

To validate the feature enhancement effect of the MSPF_Attention module, we compared the feature maps output from the 9th network layer between the original SPPF module and the improved module, analyzing their differences in background noise suppression and small-target feature focusing capabilities. Experiments were conducted using identical input images to observe the activation response distribution in feature maps (as shown in Fig 8).

Experimental results demonstrate that the MSPF_Attention module significantly reduces noise interference in background regions while enhancing local feature representation in small-target areas. Benefiting from the global contextual modeling capability of the LSKA mechanism, the model distinguishes targets from distractors in complex backgrounds through long-range dependency capture. As illustrated in Fig 8, the improved module exhibits notably weakened activation intensity of background noise in feature maps, with concentrated responses and clear boundaries in target regions, indicating enhanced model adaptability to low signal-to-noise ratio scenarios. Furthermore, detection results reveal that the original YOLO11 model suffered from missed detections and poor handling of shadow occlusion scenarios. This validates that the MSPF_Attention module effectively improves the robustness and accuracy of UAV-based vehicle small-target detection through noise suppression and target feature enhancement.

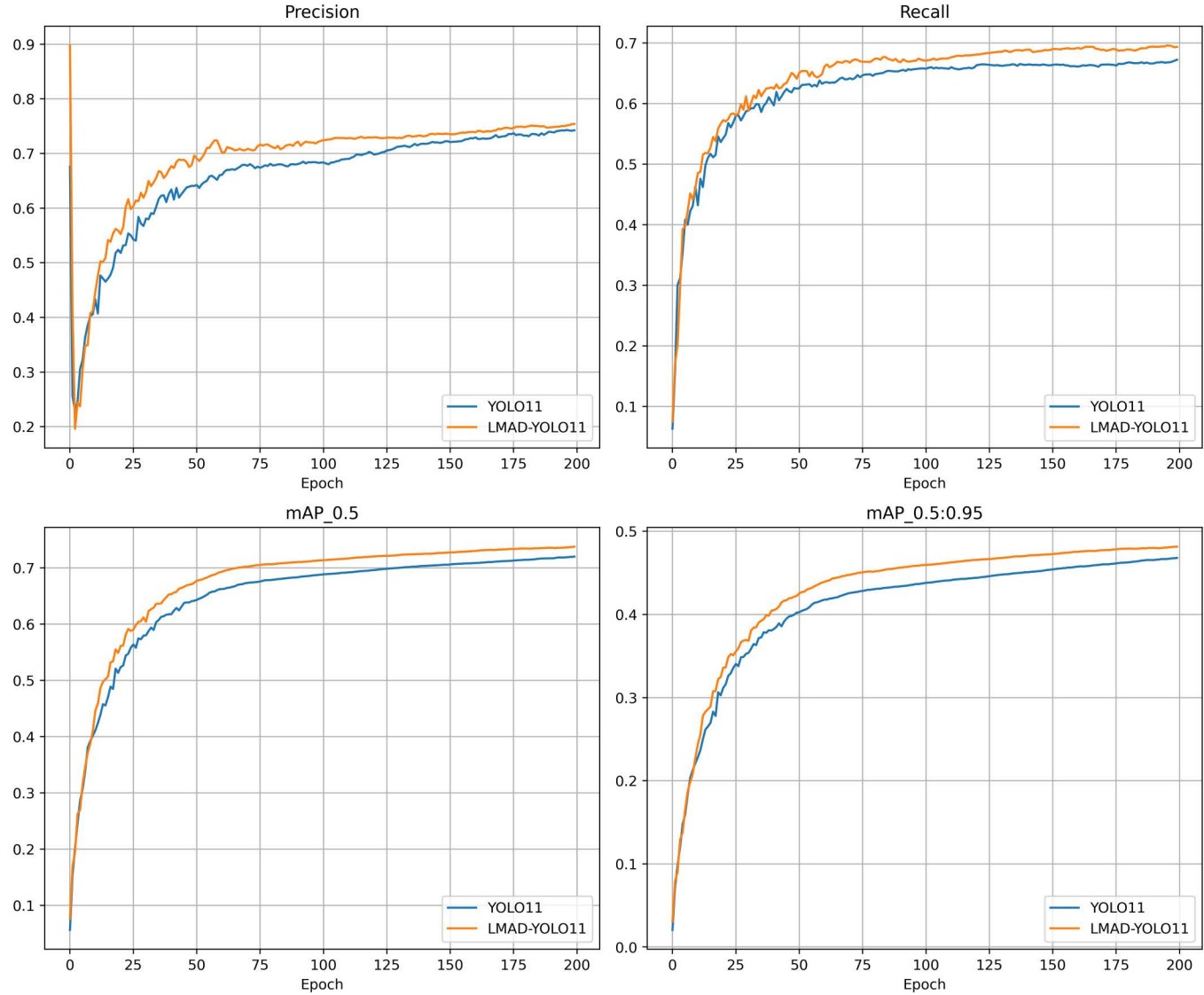

**Fig 7. Training Dynamics: Precision, Recall, and mAP Curves of LMAD-YOLO vs. YOLO11n.**

### 3.3. Receptive field comparison of improved modules

Due to the unique multi-branch structure of the MultiEdgeEnhancer module, which captures and amplifies edge feature information from different scales, we visualized and compared the receptive fields of the 3rd, 6th, and 10th layers in the LMAD-YOLO and YOLO11 models using the method from reference [23], as illustrated in Fig 9. The results demonstrate that after introducing the MultiEdgeEnhancer module and the ADown module, the receptive fields of the P4 and P5 layers are significantly enlarged.

### 3.4. Contrast experiment

In order to verify the performance of the LMAD – YOLO model in UAV small – target detection, we used models such as YOLOv3t, YOLOv8n, and YOLOv10n on the vehicle detection dataset of the Drone Vehicle using the same experimental

**Fig 8. Feature map visualization of SPPF and MSPF_Attention modules for small-target enhancement.**

**Fig 9. Comparison of two improved modules and output layer receptive fields in the header network between LMAD-YOLO and YOLO11 models.**

platform. We obtained the experimental results on the test set with the same experimental parameters. The experimental results are shown in Table 3.

YOLOv3t has a precision of 77.2%, which is better than that of all other models. However, its large number of parameters and high computational demands require high – performance computing chips. The precision of the improved LMAD – YOLO model (77%) is only 0.2% lower than that of the YOLOv3t model, but its parameters and computational requirements are much lower. The mAP@50 of the YOLOv3t model was 71.7%, which is lower than that of the LMAD – YOLO model (73.6%). The difference between the mAP@50 of YOLOv5n (68.5%) and LMAD – YOLO (71.2%) shows that LMAD – YOLO has better performance. This indicates that LMAD – YOLO can not only accurately outline small targets but also provide reliable classification results in complex UAV images.

The LMAD – YOLO model performs well in small – target detection tasks. Its high precision, low parameter count, and low computational complexity enable it to achieve better real – time performance and adaptability while maintaining high performance. Compared with other models in the YOLO series, LMAD – YOLO has obvious advantages in precision, efficiency, and robustness.

### 3.5. Model generalization ability comparison experiment

To verify the generalization ability of the improved algorithm, the LMAD-YOLO model and the comparison models were evaluated on the VisDrone2019 dataset using the same experimental environment and hyperparameters. The input image size was set to 640×640, the total number of training epochs was 200, and the training batch size was 32. The experimental results are shown in Table 4.The YOLOv3t model achieved an mAP@50 of only 18.3%, indicating significant limitations in real-world scenarios represented by the VisDrone dataset. It struggled to accurately locate targets and determine their categories in complex, dynamic environments, demonstrating poor generalization performance. In contrast,

**Table 3. Comparative experimental results.**

| Model | P/% | R/% | Map50/% | Map50-95/% | GFLOPS | Params (m) | FPS |
|---|---|---|---|---|---|---|---|
| YOLOV3t | **77.2** | 64.8 | 71.7 | 42.0 | 14.3 | 9.524 | 99.7 |
| YOLOV5n | 71.7 | 64.7 | 68.5 | 42.6 | **5.8** | 2.182 | 99.6 |
| YOLOV8n | 74.0 | 66.4 | 71.2 | 44.8 | 6.8 | 2.685 | 229.8 |
| YOLOV9t | 74.8 | 66.9 | 72.1 | 45.6 | 6.4 | **1.730** | 207.4 |
| YOLOV10n | 72.5 | 64.9 | 69.7 | 43.5 | 6.5 | 2.266 | 344.1 |
| YOLO11n | 74.6 | 66.5 | 71.4 | 45.2 | 6.3 | 2.583 | 226.0 |
| RT-DETR [24] | 76.0 | 67.2 | 67.8 | 41.4 | 42.6 | 16.756 | 64.8 |
| YOLOv12n [25] | 75.4 | 65.8 | 71.6 | 45.1 | 6.3 | 2.55 | **347.7** |
| LMAD-YOLO | 77.0 | **67.9** | **73.6** | **46.7** | 6.6 | 2.422 | 197.3 |

**Table 4. Generalization experiments on VisDrone datasets.**

| Model | P/% | R/% | Map50/% | Map50-95/% | FPS |
|---|---|---|---|---|---|
| YOLOv3t | 33.3 | 20.8 | 18.3 | 9.8 | 164.0 |
| YOLOv5n | 35.7 | 28.0 | 24.9 | 13.7 | 125.4 |
| YOLOv8n | 38.8 | 28.8 | 26.4 | 14.6 | 129.1 |
| YOLOv9t | 37.6 | 29.4 | 26.8 | **15.2** | 121.7 |
| YOLOv10n | 37.7 | **30.3** | **27.1** | 14.8 | **277.7** |
| LOYO11n | 37.6 | 28.5 | 26.0 | 14.4 | 119.9 |
| RT-DETR | **40.5** | 23.4 | 20.7 | 10.8 | 46.92 |
| YOLOv12n | 37.4 | 28.7 | 26.3 | 14.7 | 139.9 |
| LMAD-YOLO | 39.3 | 29.4 | 26.8 | 14.9 | 131.9 |

both YOLOv9t and LMAD-YOLO achieved an mAP@50 of 26.8%, tying for the lead among all compared models on this metric. This indicates that they perform robustly across various real-world shooting scenarios in the VisDrone ethicsset, achieving high accuracy in target localization and classification. They effectively generalize to diverse conditions, handle complex small-target detection environments, and demonstrate strong adaptability. The experiment further validates the superiority of the improved LMAD-YOLO model.

### 3.6. Visualization of test results

Fig 10 shows the detection effect of LMAD-YOLO model and YOLO11n model after training in the same experimental environment under different scenes. It can be seen that the improved model detects 21 car classes in the night scene,

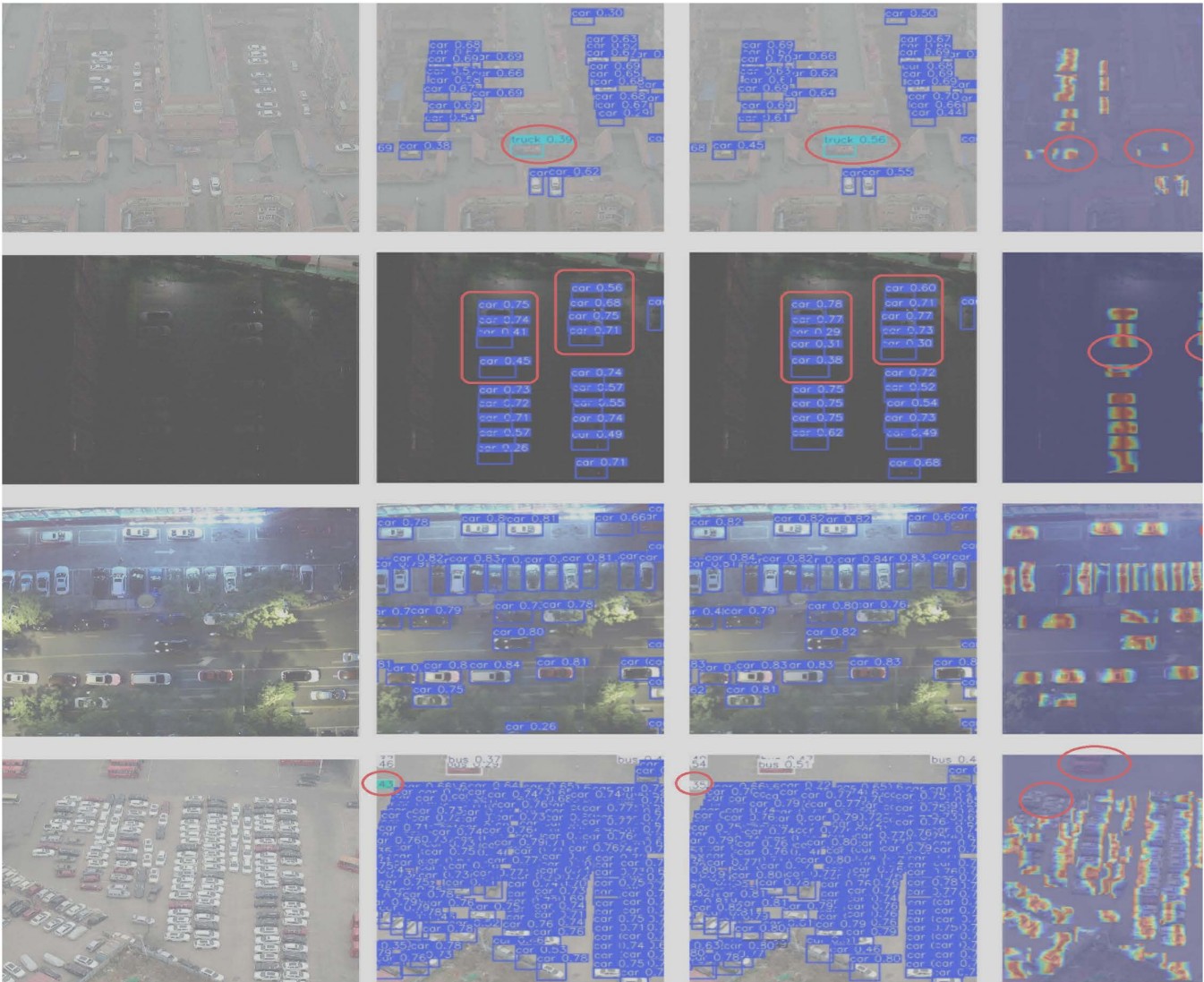

**Fig 10. Visualization of detection results on the Drone vehicle dataset (from left to right, they are the original image, YOLO11 detection image, LMAD-YOLO detection image, YOLO11 heatmap, LMAD-YOLO heatmap.** From top to bottom, they are daytime scene, nighttime scene, occlusion scene, and small target dense scene).

while yolo11n detects 20 car classes. This not only missed detection, but also the accuracy is lower than that of LMAD-YOLO model. In the dense scene, YOLO11n model identifies the bus category in the upper left corner as the truck category, while LMAD-YOLO model accurately identifies the bus category in the scene, so the improved model can avoid false detection. In other scenes, the improved model detection effect is also better than the original model. At the same time, Fig 11 shows the detection effects of LMAD-YOLO model, YOLO11n model and RT-DETR model in four different scenarios of VisDrone dataset.

## 4. Conclusions

This paper focuses on the problem of small target vehicle detection from the perspective of UAV aerial photography, deeply analyzes the difficulties of existing depth learning algorithms, and innovatively proposes the LMAD-YOLO model. Combined with multi-scale feature fusion, a MultiEdge Enhancer module is proposed to exploit the feature potential. By combining LSKA and SPPF, MSPF module reconstructs feature expression to distinguish target from interference in complex scene, which greatly improves detection reliability. Under the innovation of ADown module, the sampling path, multi-strategy parallel, reduce the parameters and computational load, improve the quality of small target features. The

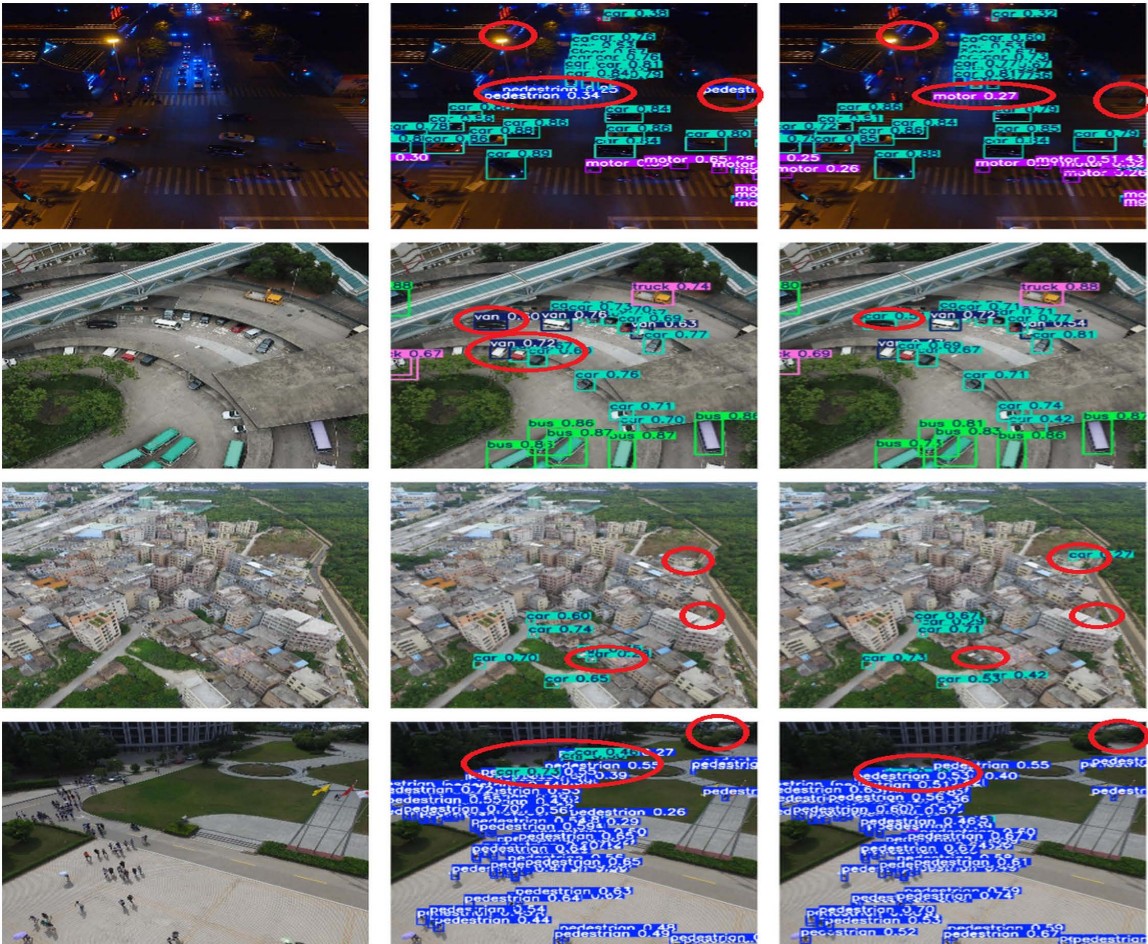

**Fig 11. Visualization of detection results on the VisDrone dataset (the columns from left to right are the original image, LMAD-YOLO detection image, YOLO11 detection image, RT-DETR detection image).**

introduction of Dasi and spread mechanism in MDFPN improves the accuracy of small target detection and reduces the risk of error detection and missing detection. The ablation, contrast, and generalization experimental system verifies the model's excellent performance. The ablation experiment confirmed the improvement value of each module. After fusion, the key indexes of the model increased significantly, and the parameters and calculation amount were controlled properly. The contrast experiment shows the superiority of precision, efficiency and robustness, the precision is better than the frontier model, the parameter and the computation are greatly optimized, the real-time and the adaptability jump, and the performance is excellent in the multi-category small target scene. In the generalization experiment, the VisDrone data set was used to test the stability of the model, adapt to the complex scene, and show the strong generalization ability.

Although the LMAD-YOLO model performs well in small target vehicle detection tasks from a drone's perspective, there are still some limitations. Firstly, the model's performance under extreme weather conditions, such as heavy rain, heavy fog, and snow, has not been fully validated. Secondly, although the Adown module reduces feature information loss, the model's receptive field may still be insufficient when dealing with extremely small targets, leading to missed detections. Future research could further optimize the model's receptive field design and combine it with super-resolution techniques to improve the detection accuracy of small targets. Additionally, introducing channel attention and self-attention mechanisms to enhance the model's ability to capture key features is also a direction for future research.

## Supporting information

**S1 File. Model training result data: includes the data generated during the model training and validation process, including various evaluation metrics, FPS, The results of each training round.**
(ZIP)

## Author contributions

**Data curation:** XiuJuan Tian.

**Formal analysis:** Le Wan.

**Funding acquisition:** XiuJuan Tian.

**Investigation:** FaHui Luo, Kang Lu.

**Methodology:** FaHui Luo, Yuqi Peng.

**Project administration:** Xue Xing.

**Resources:** Xue Xing.

**Software:** Xue Xing.

**Supervision:** Xue Xing.

**Validation:** Xue Xing, FaHui Luo, Kang Lu.

**Visualization:** FaHui Luo.

**Writing – original draft:** FaHui Luo.

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
