## [Decision Letter · Decision Letter 0]

Dear Dr. Xing,

Thank you for submitting your manuscript to PLOS ONE. After careful consideration, we feel that it has merit but does not fully meet PLOS ONE’s publication criteria as it currently stands. Therefore, we invite you to submit a revised version of the manuscript that addresses the points raised during the review process.

**ACADEMIC EDITOR: Please address all the comments and proof read the revised manuscript before resubmitting the paper.**

We look forward to receiving your revised manuscript.

Kind regards,

Qichun Zhang, PhD

Academic Editor

PLOS ONE

Journal Requirements:

This research was funded by the Jilin Province Science and Technology Development Program [YDZJ202301ZYTS291]and the Excellent Young Project of the Jilin Provincial Department of Education for the year 2024 [JJKH20240386KJ]

5. Please ensure that you refer to Figure 5, 7, 9, 10 and 11, in your text as, if accepted, production will need this reference to link the reader to the figure.

6. We note you have included a table to which you do not refer in the text of your manuscript. Please ensure that you refer to Table 3 in your text; if accepted, production will need this reference to link the reader to the Table.

7. Please remove your figures from within your manuscript file, leaving only the individual TIFF/EPS image files, uploaded separately. These will be automatically included in the reviewers’ PDF**.**

Additional Editor Comments:

Two reviewers returned the comments which are generally positive. A few details are needed to improve the quality of the paper while the motivation is needed to compare different algorithms. To enrich the background as a comparison, probably the additional reference is helpful, for example : 'A colonic polyps detection algorithm based on an improved YOLOv5s'.

Reviewers' comments:

Reviewer's Responses to Questions

**Comments to the Author**

1. Is the manuscript technically sound, and do the data support the conclusions?

Reviewer #1: Yes

Reviewer #2: Partly

2. Has the statistical analysis been performed appropriately and rigorously?

Reviewer #1: Yes

Reviewer #2: No

3. Have the authors made all data underlying the findings in their manuscript fully available?

Reviewer #1: Yes

Reviewer #2: Yes

4. Is the manuscript presented in an intelligible fashion and written in standard English?

Reviewer #1: Yes

Reviewer #2: No

Reviewer #1: This paper introduces LMAD-YOLO, a vehicle image detection algorithm aimed at addressing the challenges of small target vehicle detection in drone aerial photography. The proposed method incorporates several improvements, including the MultiEdgeEnhancer module, MSPF module, Adown module, and the MDFPN network, claiming performance gains on the DroneVehicle and VisDrone datasets. While acknowledging the innovative aspects, the paper requires further refinement in its methodology, experimental validation, results analysis, and overall presentation.

Introduction:

The introduction provides a somewhat general overview of the challenges in small target vehicle detection from drones, lacking an in-depth analysis of the specific limitations of existing mainstream methods in addressing these challenges.

The review of related work appears insufficient in scope and depth, failing to clearly establish the novelty and necessity of the presented work within the existing literature.

The specific goals and the precise problems the research intends to solve are not articulated clearly enough, leaving the reader with a somewhat vague understanding of the study's motivation.

Methodology:

The principles and design rationale behind the core modules (MultiEdgeEnhancer, MSPF, Adown, and MDFPN) are not described with sufficient clarity and depth. Explanations of some key formulas or operations are too brief, making it difficult for the reader to fully grasp their effectiveness.

For instance, the specific implementation and underlying principles of "adaptive Avgpool2d module for multi-scale aggregation," how EdgeBooster enhances edge information through differential calculations and convolution activation, and the detailed structure and operation of LSKA within the MSPF module are not adequately elaborated.

The overall network architecture diagram (Figure 1) lacks sufficient detail, such as the connections between modules and the changes in feature map channel numbers, which would aid in understanding the network's structure and data flow.

Experiments:

Dataset Description: While the DroneVehicle and VisDrone datasets are mentioned, the paper lacks detailed statistical information about them, such as the number of samples per class and the distribution of object sizes, which are crucial for an objective evaluation of the experimental results.

Baseline Selection: The choice of baseline models for comparison, while including some YOLO series variants, omits comparisons with recent state-of-the-art methods in drone-based small object detection, limiting the assessment of the proposed method's superiority.

Ablation Study Analysis: The analysis of the ablation study results in Table 2 primarily focuses on performance metric changes, lacking a deeper investigation into how different module combinations affect model performance and potential interactions between modules.

Results and Discussion:

The presentation of results relies heavily on tabular data, with insufficient in-depth analysis and interpretation of the experimental findings.

The discussion section does not adequately leverage the experimental results to provide a thorough exploration of the proposed method's strengths and limitations, nor does it offer a detailed comparative analysis with existing works.

Visual results in Figures 8, 10, and 11, while providing some qualitative evidence, are limited in quantity and lack detailed interpretation to clearly demonstrate the specific improvements brought by the proposed modules across various scenarios and challenges.

Presentation:

The logical flow within certain paragraphs could be improved, and the language used is not consistently fluid and precise.

The captions and legends of figures and tables lack sufficient information to clearly convey the intended message. For example, Figure 7 lacks detailed explanations of the axes and legends. The titles of Figures 8 and 9 are too simplistic and do not adequately describe the visualized content.

The formatting of the references may exhibit inconsistencies and requires careful review to ensure uniformity and completeness.

Reviewer #2: In my opinions, the manuscript has the following deficiencies which cannot be ignored. Language and presentation, should be refined throughout the text, its stylistic and grammatical issues, significantly hinders the readability of the work. Why is the YOLO11n chosen to be improved, there are different editions of YOLO11 models, such as YOLO11m, YOLO11s, YOLO11x, the experimental design is limited. Additionally, more interesting test data, such as evaluation indices, FPS, mAP@0.5:0.95, should be added.

**Do you want your identity to be public for this peer review?** For information about this choice, including consent withdrawal, please see our Privacy Policy

Reviewer #1: No

Reviewer #2: No

---

## [Author Response · Author response to Decision Letter 1]

8 May 2025

Comment reply in the "Response to Reviewers. pdf" file

---

## [Decision Letter · Decision Letter 1]

LMAD-YOLO: A Vehicle Image Detection Algorithm for Drone Aerial Photography Based on Multi-Scale Feature Fusion

PONE-D-25-12249R1

Dear Dr. Xing,

We’re pleased to inform you that your manuscript has been judged scientifically suitable for publication and will be formally accepted for publication once it meets all outstanding technical requirements.

Kind regards,

Qichun Zhang, PhD

Academic Editor

PLOS ONE

Additional Editor Comments:

Reviewer 2 has not accepted the re-invitation for the revised version. Thus, I, academic editor, has reviewed the revision as the additional reviewer. Basically, all the concerns have been addressed well and I recommend accepting this submission.

Reviewers' comments:

Reviewer's Responses to Questions

**Comments to the Author**

Reviewer #1: All comments have been addressed

2. Is the manuscript technically sound, and do the data support the conclusions?

Reviewer #1: Yes

3. Has the statistical analysis been performed appropriately and rigorously?

Reviewer #1: Yes

4. Have the authors made all data underlying the findings in their manuscript fully available?

Reviewer #1: Yes

5. Is the manuscript presented in an intelligible fashion and written in standard English?

Reviewer #1: Yes

Reviewer #1: I strongly suggest the authors further refine the language throughout the manuscript. While the overall clarity is acceptable, the writing can be more concise and polished. Eliminating redundancy and tightening the expression would significantly enhance the readability and impact of the work.

**Do you want your identity to be public for this peer review?** For information about this choice, including consent withdrawal, please see our Privacy Policy

Reviewer #1: No

---

## [Editor Report · Acceptance letter]

PONE-D-25-12249R1

PLOS ONE

Dear Dr. Xing,

I'm pleased to inform you that your manuscript has been deemed suitable for publication in PLOS ONE. Congratulations! Your manuscript is now being handed over to our production team.

Kind regards,

on behalf of

Prof. Qichun Zhang

Academic Editor

PLOS ONE